Peanut-based intercropping systems altered soil bacterial communities, potential functions, and crop yield

Liu Zhu 1 2
Nan Zhenwu 1
Lin Songming 1 3
Meng Weiwei 1
Xie Liyong 2
Yu Haiqiu 2
Zhang Zheng zhangzheng7268@163.com 1
Wan Shubo wanshubo2016@163.com wansb@saas.ac.cn 1
1 Shandong Academy of Agricultural Sciences , Ji’nan , China
2 College of Agronomy, Shenyang Agricultural University , Shenyang , China
3 Qilu Normal University , Ji’nan , China
Thomas Jonathan
Electronic publication date: 2024 Feb 7
Publication date: 2024
Volume: 12
Electronic Location ID: e16907
Received 2023 Sep 5; Accepted 2024 Jan 17
Copyright: ©2024 Liu et al.
Copyright year: 2024
Copyright holder: Liu et al.
License: This is an open access article distributed under the terms of the Creative Commons Attribution License, which permits unrestricted use, distribution, reproduction and adaptation in any medium and for any purpose provided that it is properly attributed. For attribution, the original author(s), title, publication source (PeerJ) and either DOI or URL of the article must be cited.
License URL: https://creativecommons.org/licenses/by/4.0/

Keywords: Sorghum, Millet, Peanut, Intercropping, Yield, Soil bacterial community

Funding: Key Research and Development Program of Shandong Province, China 2021CXGC010804 National Key Research and Development Program of China 2020YFD1000905 National Modern Agricultural Industrial Technology System Funding Project CARS-13 This work was supported by the Key Research and Development Program of Shandong Province, China (2021CXGC010804), the National Key Research and Development Program of China (2020YFD1000905) and the National Modern Agricultural Industrial Technology System Funding Project (CARS-13). The funders had no role in study design, data collection and analysis, decision to publish, or preparation of the manuscript.

==============================
Intercropping is an efficient land use and sustainable agricultural practice widely adopted worldwide. However, how intercropping influences the structure and function of soil bacterial communities is not fully understood. Here, the effects of five cropping systems (sole sorghum, sole millet, sole peanut, sorghum/peanut intercropping, and millet/peanut intercropping) on soil bacterial community structure and function were investigated using Illumina MiSeq sequencing. The results showed that integrating peanut into intercropping systems increased soil available nitrogen (AN) and total nitrogen (TN) content. The alpha diversity index, including Shannon and Chao1 indices, did not differ between the five cropping systems. Non-metric multidimensional scaling (NMDS) and analysis of similarities (ANOSIM) illustrated a distinct separation in soil microbial communities among five cropping systems. Bacterial phyla, including Actinobacteria, Proteobacteria, Acidobacteria, and Chloroflexi, were dominant across all cropping systems. Sorghum/peanut intercropping enhanced the relative abundance of phyla Actinobacteriota and Chloroflexi compared to the corresponding monocultures. Millet/peanut intercropping increased the relative abundance of Proteobacteria, Acidobacteriota, and Nitrospirota. The redundancy analysis (RDA) indicated that bacterial community structures were primarily shaped by soil organic carbon (SOC). The land equivalent ratio (LER) values for the two intercropping systems were all greater than one. Partial least squares path modeling analysis (PLS-PM) showed that soil bacterial community had a direct effect on yield and indirectly affected yield by altering soil properties. Our findings demonstrated that different intercropping systems formed different bacterial community structures despite sharing the same climate, reflecting changes in soil ecosystems caused by interspecific interactions. These results will provide a theoretical basis for understanding the microbial communities of peanut-based intercropping and guide agricultural practice.

Introduction

In modern agricultural practices, crop production tends towards monoculture. However, continuous monocropping leads to the progressive loss of soil quality, resulting in a decline in yield and quality, as well as the prevalence of soil-borne diseases, known as replanting obstacle (Chen et al., 2020; Luo et al., 2023). Therefore, it is important to adopt strategies to overcome replanting obstacle. Among diverse agricultural practices, intercropping has captured more and more attention due to its agro-ecological advantages (Bardgett & Van der Putten, 2014; Pang et al., 2022; Wang et al., 2022). There are many benefits of intercropping, such as increasing yield (Carton et al., 2020), improving the utilization efficiency of nutrient resources (Fan et al., 2020), increasing the biodiversity and stability of farmland, continuously controlling diseases, pests, and weeds, and improving the ecological environment of farmland (Chen, 2000). Intercropping has been demonstrated to alleviate the obstacles caused by continuous monocropping (Han et al., 2022). Peanut (Arachis hypogaea L.) is an essential oilseed and protein-producing crop. Integrating peanut into intercropping systems offers a potential solution to reduce N fertilizer input and enhance N cycling (Erhunmwunse et al., 2023). At the same time, the agricultural benefits of intercropping systems can be utilized to alleviate the replanting obstacle of peanut. However, how peanut-based intercropping affects the structure and function of soil bacterial communities has yet to be fully understood.

Soil microorganism plays a key role in the soil ecosystem functions, such as nutrient cycle, energy flow, and organic matter decomposition (Van der Heijden & Wagg, 2013; Khan et al., 2023). Soil bacteria account for 70%–90% of the total soil microbial biomass, and its physiological groups play a crucial role in soil conservation, plant growth, and crop quality (Vuyyuru et al., 2020; Aguilera-Huertas et al., 2023). Land use patterns, cropping systems, straw returning, and vegetation types can affect the species and quantity of soil bacteria and the spatial distribution of the soil bacterial community (Mayer et al., 2019; Bu et al., 2020; Yang et al., 2020; Nan et al., 2021). Several studies have illustrated that intercropping systems considerably impact the composition of soil microbial communities (Wang et al., 2015; Cao et al., 2017), such as increased abundance of nitrogen fixation-associated bacteria in maize/peanut intercropping systems (Chen et al., 2018). Moreover, research has revealed that cereal/peanut intercropping significantly enhanced the biomass of soil bacteria (Li et al., 2016). The soil quality was improved by increasing the abundance of beneficial bacteria in the cassava and peanut intercropping system (Tang et al., 2020). Similarly, sugarcane/peanut intercropping systems can improve soil quality and increase the abundance of beneficial microbes (Tang et al., 2021). Zhao et al. (2022) demonstrated that maize/peanut intercropping increased the functional diversity of bacterial communities.

Peanut intercropping with millet or sorghum is a typical cereal/legume intercropping. It can improve yields through the use of soil resources. However, more studies need to understand the influences of intercropping on soil microbial communities. Therefore, we explored the responses of the microecological soil environment in two peanut-based intercropping systems (sorghum/peanut intercropping and millet/peanut intercropping) using 16S amplicon sequencing. The aims of this study were (i) to analyze the characteristics of soil properties, bacterial communities, and crop yields in different cropping systems; (ii) to explore the interaction relationships of soil properties and bacterial communities; (iii) to quantify their impact on crop yields.

Materials & Methods

Experimental site and design

The test site was in 2019 at Anjiazhuang Town, Feicheng City, Shandong Province, China (116°46′E, 35°57′N). The site has a continental monsoon climate with four distinctive seasons, sufficient light, and a warm climate. The mean annual temperature was 12.9 °C and the annual rainfall was 660 mm. The frost-free period was 200 d. And the annual sunshine hours totaled 2607 h. Physicochemical properties of the surface soil layer (0–20 cm) were the following: organic matter, 11.6 g kg−1; total nitrogen, 1.0 g kg−1; alkaline nitrogen, 75.9 mg kg−1; available phosphorus, 28.0 mg kg−1; available potassium, 168.6 mg kg−1; pH, 7.0. The climate data was collected from a small, self-built weather station.

The field experiment was carried out using a randomized complete block design. In this experiment, sole millet (Setaria italica, labeled Millet), sole sorghum (Sorghum bicolor, labeled Sorghum), and sole peanut (Arachis hypogaea, labeled Peanut), and two intercropping patterns of sorghum/peanut intercropping of 3:4 (Fig. S1A, labeled Sorg/Pean) and millet/peanut intercropping of 4:4 (Fig. S1B, labeled Mill/Pean) were set. The field area of each plot was 0.45 ha, with three plot replicates per treatment. The tested cultivars of millet, sorghum, and peanut were “Jigu 20”, “Jiliang 3”, and “Huayu 36”, respectively. The compound fertilizer (N/P2O5/K2O = 15:15:15) of 750 kg ha−1 was broadcast on the field as basal fertilizer before sowing. The millet, sorghum and peanut are all sown simultaneously in mid-May. Millet and sorghum are harvested in early September, and peanuts in mid-September. The field management was implemented according to the practices of local farmers.

Sample collection and indices calculation

The soil samples were obtained during the harvest period of crops on 12 September 2019. A soil sampler was used to collect 0–20 cm of soil from each plot. Soil sampling was carried out at five points following the five-point sampling method. These soil samples were then blended into one sample. Thus, a total of 15 samples (five treatments with three replicates) were obtained.

After removing the impurities such as sand, stone, and root, fresh soil samples were transferred into aseptic cryopreservation tubes and stored in a −80 °C refrigerator to analyze soil bacterial community structure. Moreover, the remaining samples were dried naturally and used to determine physical and chemical properties.

At maturity, two ridges × 4 m (3.2 m2), two rows × 4 m (6.4 m2), and three rows × 4 m (7.2 m2) per plot were randomly harvested to examine the yields of peanut, millet, and sorghum, respectively.

The land equivalent ratio (LER) was calculated as follows (Mead & Willey, 1980): (1.1) LER=pLERA+pLERB=YiAYsB+YiBYsB

where pLERA and pLERB indicate partial LER for crop A and crop B, YiA and YsB are the yield of crop A in intercropping and sole-cropping, and YiB and YsB are the yield of crop B in intercropping and sole-cropping, respectively. The LER value >1 indicates that intercropping system has yield advantage.

Soil physical and chemical properties

The total nitrogen (TN) content was measured using the micro-Kjeldahl procedure (Cao et al., 2017). The available nitrogen (AN), available phosphorus (AP), and available potassium (AK) content were measured by alkaline diffusion, NaHCO3 leaching molybdenum-antimony reverse absorption spectrophotometry and ammonium acetate atomic absorption spectrophotometry, respectively (Cao et al., 2017). The soil organic carbon (SOC) was measured by potassium dichromate oxidation (Wang et al., 2015). And the pH was determined in a suspension with a soil: water ratio of 1:2.5 according to the potentiometric method (Li et al., 2016).

Soil DNA extraction, PCR amplification and sequencing

Total DNA was extracted from the tissue using E.Z.N.A.® soil DNA kit (Omega Bio-tek, Norcross, GA, US) according to the manufacturer’s instructions. The DNA quality was examined by 1% agarose gel electrophoresis. The concentration and purity of extracted DNA were determined using the NanoDrop2000.

The primer set of V338f (5′-ACTCCTACGGGAGGCAGCAG- 3′) and V806R (5′-GGACTACHVGGGTWTCTAAT-3′) was used to amplify the V3–V4 region of the bacterial 16S rRNA (Tang et al., 2020). The PCR products from the same sample were mixed and recovered on a 2% agarose gel. The recovered products were purified using the AxyPrep DNA Gel Extraction Kit (Axygen Biosciences, Union City, CA, USA). The recovered products were detected by 2% agarose gel electrophoresis and quantified by Quantus™ Fluorometer (Promega, Madison, WI, USA). The sequencing library was constructed using the NEXTFLEX Rapid DNA-Seq Kit following manufacturer’s recommendations. Sequencing libraries were mixed with PhiX. Reads were removed during data demultiplexing (Demultiplexing refers to splitting multiplexed reads from different or the same lane based on the index, generates the sample’s corresponding fastq file). The Shanghai Majorbio Bio-pharm Biotechnology Co., Ltd. (Shanghai, China) Illumina Miseq PE300 platform was used for high-throughput sequencing. Negative controls were used when performing amplification and were not used during sequencing.

Sequence analysis

The paired-end data from the sequencing were demultiplexed according to the barcode to remove the adapters and barcodes (each barcode was only used for a single sample). The adapter sequences at the 3′ and 5′ ends of the reads were quality clipped using the FASTP version 0.20.0 (https://github.com/OpenGene/fastp). A standard procedure was performed for quality control using DADA2 in Qiime2 version 2019.7.0 (https://qiime2.org/) to obtain the amplicon sequence variants (ASVs) (Bolyen et al., 2019; Wang et al., 2023). We removed any sequences having (i) an average base mass <20, (ii) a length <50, and (iii) N bases. The DADA2 plugin in QIIME2 was used for sequence quality trimming, denoising, merging, and chimera detection (Callahan et al., 2016). The consensus sequences for the ASVs were classified with a Naive Bayes classifier trained against the SILVA 16S rRNA gene reference (release 132) database.

Statistical and bioinformatics analysis

Statistical analysis was carried out using R (version 3.3.1; R Core Team, 2016). Alpha diversity indices (Shannon and Chao1 indices) were calculated by Mothur (version 1.30.). Nonmetric multidimensional scaling (NMDS) and analysis of similarity (ANOSIM) based on the Bray–Curtis similarity matrix were used to test the differences in the compositions of the bacterial communities among different cropping systems. Linear discriminant analysis (LDA) effect size (LEfSe), according to the cutoff value of LDA >3.5, was used to identify certain specific biomarkers between different groups (Segata et al., 2011). Redundancy analysis (RDA) was carried out to investigate the effect of soil physicochemical properties on bacterial abundance at the bacterial phylum and genus levels. The data were analyzed using SPSS 20 (IBM, USA). One-way analysis of variance (ANOVA) and least significance difference (LSD) test were performed to determine the differences of soil factors and microbial diversity in different cropping treatemnts.

Results

Grain yield and LER

The total grain yield in the grain sorghum/peanut intercropping system was 8075 kg ha−1, which increased by 29.2% and 10.3%, compared with the Peanut and Sorghum, respectively (Fig. 1A). The yield of Mill/Pean was slightly higher than Peanut but significantly higher than Millet. Additionally, the LER values of two intercropping systems were all greater than one, and the higher LER value was obtained in Sorg/Pean (Fig. 1B).

Soil physicochemical properties

Soil physicochemical properties of all samples are shown in Table 1. Intercropping practices changed the soil characteristics. Compared with sole millet, intercropping with peanut significantly increased the AN and TN content by 9.9% and 4.9%, respectively (p < 0.05). Compared with sole sorghum, intercropping with peanut significantly increased the AN and TN content by 16.0% and 10.3%, respectively (p < 0.05). The pH showed no significant difference among the five cropping systems. The AP was positively correlation with SOC, while the SOC, AP, and AK were negatively correlated with TN (Fig. S2).

Bacterial community diversity

A total of 4157 ASVs were detected in all samples. The number of sequencing reads after denoising and removing chimeras for each sample was shown in Fig. S3. Venn diagrams of observed ASVs in each treatment are shown in Fig. S4. The five treatments shared 683 ASVs (17%). Each treatment has its own unique ASVs. Sorghum had the most (23%), whereas Sorg/Pean had the least (16%). There were no statistically significant differences in bacterial community richness (Chao1) or alpha diversity indices (Shannon) among Sorghum, Millet, Peanut, Sorg/Pean, and Mill/Pean (Fig. S5).

Nonmetric multidimensional scaling (NMDS) analysis was carried out using the Bray–Curtis dissimilarities to reveal the soil bacteria beta diversity of five cropping systems. The results showed that the taxa of soil bacterial communities were distinct among five cropping systems (Stress = 0.04, Fig. 2A). Furthermore, the analysis of similarities (ANOSIM) also confirmed that the bacterial community differed among five cropping systems ( R = 0.858, p = 0.001, Fig. 2B).

Figure 1 The grain yield (A) and land equivalent ratio (LER, B) of all treatments.

Same lowercase letters indicate non-significant differences at p < 0.05 level.

Table 1 Physical and chemical properties of the five cropping systems.

Treatment	TN (g kg−1 )	SOC (g kg−1 )	AN (mg kg−1 )	AP (mg kg−1 )	AK (mg kg−1 )	pH	
Sorghum	0.97d	7.52a	71.8b	49.78a	175.21a	6.97a	
Millet	1.02c	7.43ab	75.52b	46.09ab	176.86a	7.04a	
Peanut	1.16a	7.30b	77.72b	40.81b	165.25b	7.02a	
Sorg/Pean	1.07b	7.37ab	83.3a	45.08ab	172.77ab	7.01a	
Mill/Pean	1.07b	7.35b	83.02a	40.49b	170.54ab	7.00a	
Notes.

TN total nitrogen

SOC soil organic carbon

AN available nitrogen

AP available phosphorus

AK available potassium

Same lowercase letters indicate non-significant differences at p < 0.05 level.

Figure 2 Beta diversity of soil bacterial community under five cropping systems.

(A) Non-metric multidimensional scaling (NMDS) analysis. Dots with diverse shapes represent samples of five cropping systems. (B) Analysis of similarity (ANOSIM). Between reflects the differences between groups, and groups Sorghum, Millet, Peanut, Sorg/Pean, and Mill/Pean represent the differences within groups.

Bacterial community structure

The taxonomic distribution of the phylum and the genus is presented in Fig. 3. At the phylum level, all treatments possessed similar dominant phyla of microbial communities. Actinobacteria (27%), Proteobacteria (23%), Acidobacteria (20%), and Chloroflexi (10%) were the most abundant phyla, followed by Firmicutes (3.8%), Gemmatimonadota (3.1%), Bacteroidota (3.0%) and Myxococcota (2.5%). Methylomirabilota and Nitrospirae were detected at low relative abundances (<2.0%). However, the abundance of dominant phyla varied among treatments. Sorg/Pean enhanced the relative abundance of phyla Actinobacteriota and Chloroflexi while reducing the relative abundance of Acidobacteriota, Bacteroidota, and Myxococcota compared to the corresponding monocultures. Mill/Pean enhanced the relative abundance of Proteobacteria, Acidobacteriota, and Nitrospirota while decreasing the relative abundance of Chloroflexi and Firmicutes. At the genus level, the relative abundances of the top ten dominant genera accounted for 17.4% of total sample sequences. The top ten genera were Sphingomonas (3.0%), RB41 (2.8%), Bacillus (2.5%), Gaiella (2.0%), Nocardioides (1.4%), Microvirga (1.3%), Streptomyces (1.3%), MND1 (1.2%), Nitrospira (1.2%) and Lysobacter (0.8%).

Figure 3 Percentage of the top ten most abundant bacterial groups at phylum (A) and genus (B) levels.

Taxonomic composition

Linear discriminant analysis (LDA) effect size (LEfSe) showed biomarkers of soil bacterial lineages in five cropping systems (Fig. 4). Cropping systems specifically affected the bacterial community from the phylum to the genus level. For example, Sorg/Pean had high abundances of Streptomyces and Microvirga. Mill/Pean soil increased the proportions of Kribbella and Reyranella. Candidatus_Solibacter, Nitrospira, and Sphingomonas were significantly enriched in Sorghum soil. Enrichment of Nocardioides and Gaiella was significant in Millet soil. In addition, Rubrobacter and Pontibacter were enriched in Peanut soil.

Figure 4 Differences in classification levels of bacterial community from phylum to genus.

The threshold of LDA score was 3.5. The cladogram consists of phylum, class, order, family, and genus levels in order from inside to outside. Yellow nodes represent the taxa that did not play an important role in any of the treatment groups, and other colored nodes represent taxa that play an important role in the corresponding color group.

Correlation between soil bacterial groups and physicochemical properties

Redundancy analysis (RDA) (Fig. 5) and Pearson correlation analysis (Table 2) were carried out to determine the relation of the soil factors to microbial communities in soils. The pH and TN were positively correlated with each other, and the SOC, AK, and AP were positively correlated with each other. The SOC had the most significant impact on bacterial community composition. Other soil properties such as AN, AP, and TN were also important factors affecting the bacterial community structure.

Figure 5 Redundancy analysis (RDA) of the relationship between bacterial community structure and soil chemical properties at phylum (A) and genus (B) levels.

TN, total nitrogen; SOC, soil organic carbon; AN, available nitrogen; AP, available phosphorus; AK, available potassium.

Table 2 Pearson correlation analysis between the relative abundances of dominant bacteria taxa and soil physicochemical parameters.

Bacterial Taxa	TN	SOC	AN	AP	AK	pH	
Phylum	Actinobacteriota	0.190	−0.556*	0.529*	−0.325	−0.017	0.125	
	Proteobacteria	−0.476	0.618*	−0.531*	0.564*	0.257	−0.189	
	Acidobacteriota	0.217	0.120	−0.006	−0.200	−0.206	−0.047	
	Chloroflexi	0.291	−0.624*	0.291	−0.392	−0.280	0.228	
	Firmicutes	0.183	−0.417	0.139	−0.077	−0.118	−0.175	
	Gemmatimonadota	−0.397	0.562*	−0.402	0.465	0.339	0.144	
	Bacteroidota	−0.366	0.644**	−0.690**	0.398	0.274	0.017	
	Myxococcota	0.242	−0.097	0.001	−0.183	−0.233	0.124	
	Methylomirabilota	0.393	−0.275	0.242	−0.064	−0.189	0.119	
	Nitrospirota	−0.630*	0.664**	−0.465	0.574*	0.329	−0.315	
Genus	Sphingomonas	−0.524*	0.727**	−0.562*	0.632*	0.338	−0.082	
	RB41	0.030	0.437	−0.492	0.257	−0.077	0.203	
	Bacillus	0.298	−0.410**	0.025	−0.072	−0.208	−0.137	
	Gaiella	0.445	−0.655	0.330	−0.392	−0.183	0.239	
	Nocardioides	0.070	−0.330	0.027	−0.122	0.166	0.213	
	Microvirga	0.115	−0.134	−0.025	0.176	0.094	0.075	
	Streptomyces	−0.001	−0.487	0.468	−0.278	0.027	0.023	
	MND1	−0.049	0.403	−0.620*	0.222	−0.035	0.173	
	Nitrospira	−0.630*	0.664**	−0.465	0.574*	0.329	−0.315	
	Lysobacter	−0.462	0.668**	−0.621*	0.575*	0.432	0.168	
Notes.

TN total nitrogen

SOC soil organic carbon

AN available nitrogen

AP available phosphorus

AK available potassium

* p < 0.05.

** p < 0.01.

*** p < 0.001.

At the phylum level, Actinobacteriota, Proteobacteria, Chloroflexi, Gemmatimonadota, Bacteroidota, and Nitrospirota among the top ten bacterial phyla were highly correlated with SOC. At the genus level, genera Sphingomonas, Nitrospira, and Lysobacter showed a positive correlation with SOC, while the opposite result was observed with the Bacillus.

Potential metabolic pathways

To understand the potential function of the soil microbial community, metabolic properties and pathways in different cropping systems were predicted by Tax4fun. Based on the Kyoto Encyclopedia of Genes and Genomes (KEGG), 7834 KEGG orthologues (KO) were found across all samples, mainly belonging to metabolism, genetic information processing, environmental information processing, and environmental information processing, cellular processes, and human diseases. In second hierarchy level bacterial functions, the top abundant functional pathways (relative abundance >1%) are shown in Fig. S6. Ten pathways were observed for metabolism, three for genetic information processing, two for environmental information processing, two for cellular processes, and one for human diseases. Compared to other pathways, carbohydrate metabolism, amino acid metabolism, and membrane transport were overrepresented.

PLS-PM analysis

The result of partial least squares path modeling analysis (PLS-PM) showed that beta diversity and soil properties had significant impacts on yield (Fig. 6). Among them, soil bacterial community (beta diversity, phylum abundance, and genus abundance) had a direct impact on yield and indirectly influenced yield via changing soil properties.

Discussion

Agricultural practices, including intercropping and rotation, can promote soil health (Zou et al., 2021). Intercropping, especially cereal/legume intercropping, plays a crucial role in enhancing soil nutrient levels (Liu et al., 2014b; Li & Wu, 2018; Zhang et al., 2019). Previous studies confirmed that maize/peanut intercropping improved the contents of total N, available N, total P, available P, total K, and available K (Han et al., 2022). In our study, the two intercropping systems (millet/peanut intercropping and sorghum/peanut intercropping) had little effect on soil physicochemical properties(including pH, SOC, AP, and AK), but soil AN and TN contents were remarkably increased in the intercropping system (Table 1). These results might be attributed to the biological nitrogen fixation of peanut, which increased nitrogen content in soil (Liu et al., 2023). In addition, intercropped legumes could fix more nitrogen from the atmosphere than monocultures due to the competition for soil nitrogen from cereals (Cowden et al., 2020; Rodriguez et al., 2020).

Figure 6 Partial least squares path model (PLS-PM) for the yield.

Red and blcak arrows indicate positive and negative relationships, respectively. *p < 0.05, **p < 0.01, ***p < 0.001.

In intercropping systems, the diversity and richness of the bacterial community did not differ from sole cropping (Table 2). These results were in line with the research results in wheat/pea intercropping (Pivato et al., 2021) and maize/peanut intercropping (Zhao et al., 2022). However, studies in sugarcane/soybean intercropping found that intercropping significantly increased bacterial diversity (Liu et al., 2021). The variance in microbial diversity might be relevant to the competitiveness of resident microorganisms (Goerges et al., 2008). The changes in resident microbial communities may lead to alterations in ecosystem functions, such as soil nutrient cycling and decomposition, by mediating soil enzyme activities (Zhao et al., 2021). The competition from resident microorganisms may greatly increase the number of rare communities, thereby altering the structure of the soil microbial community (Mawarda et al., 2020).

Microbial community composition in soils is influenced by various factors, including environmental parameters, fertilizer application, pesticides, and agricultural practices (Chamkhi et al., 2022). Our findings indicated that the bacterial community composition was influenced by cropping systems. The dominant phyla, Actinobacteria, Proteobacteria, Acidobacteria, and Chloroflexi, were consistent across all cropping systems, indicating that intercropping did not alter the dominant phylum. In contrast to the results for Mill/Pean, Sorg/Pean increased the relative abundance of Chloroflexi and decreased the relative abundance of Acidobacteriota, demonstrating that intercropping of different combinations changes the abundance of many microbial species. The main reason for these differences is that differences in intercropping crop cover, residues, and root secretions can have an impact on the soil environment, which in turn alters the structure and functional activity of the soil microbial community (Broeckling et al., 2008; Liu et al., 2022). In addition, competition or complementarity for resources in intercropping can also affect soil microbial (Duchene, Vian & Celette, 2017). These results were consistent with previous studies that intercropping altered bacterial community abundance (Liu et al., 2014a; Zeng et al., 2020).

Crop species and interspecific interactions could also influence the composition of the soil microbial community (Wieland, Neumann & Backhaus, 2001; Li & Wu, 2018). According to a previous study based on cassava/peanut intercropping (Tang et al., 2020), the percentage of phylum Gemmatimonadetes in intercropping systems was higher than that in the sole-cropping systems. At the same time, other phyla (including Proteobacteria, Actinobacteria, Acidobacteria, and Chloroflexi) showed no significant difference between the intercropping and sole-cropping systems. These results are inconsistent with our research. The reason might be that interactions among species affected the dominant bacterial phyla (Zhang et al., 2018). Intercropping in our study promoted bacterial genera that are potentially involved in N fixation and cycling activities. For example, there was an increase in the relative abundance of Microvirga (Shahrajabian, Sun & Cheng, 2021), known for their roles in N fixation and producing phytohormones and enzymes, under Sorg/Pean soils. Similarly, we found an increase in Kribbella and Reyranella under Mill/Pean soils. Recently, Chen et al. (2022) reported that intercropped maize boosted peanut growth and yield through inducing more colonization of genera Streptomyces in peanut roots. Our results found that Streptomyces showed higher abundance under the Sorg/Pean soils. The genus Streptomyces has already been described as diazotrophic bacteria, which plays an important role in N fixation (Solanki et al., 2019). Moreover, as common rhizosphere growth-promoting bacteria (PGPR), Streptomyces can also promote plant growth and inhibit the occurrence of soil-borne disease (Sakineh et al., 2021). Thus, the increase in yield in Sorg/Pean may be related to the plant growth-promoting activity of Streptomyces.

Many edaphic factors, like soil physicochemical properties, indirectly influence the functional structure of microbial communities in natural ecosystems (Wang et al., 2015; Zhang et al., 2020b). Soil physicochemical properties, such as SOC and C/N ratio, are regarded as the driving force of soil microbial diversity (Thomson et al., 2015). Soil organic carbon (SOC) is an essential element of soil ecosystems and has important effects on soil ecosystem functions (Hu et al., 2022). In this study, we found that SOC was the primary factor influencing the soil bacterial community structure according to RDA, in line with the findings noted by Sul et al. (2013) and You et al. (2014). Soil bacterial communities are sensitive to SOC change. The reason may be that SOC is the most important part of the soil carbon pool, and carbon is a source of nutrients and energy (Huo et al., 2018). This study showed that Sphingomonas was positively correlated with SOC, which was also one of the dominant genera in the soil of Sorghum, further demonstrating that SOC could structure the composition of the bacterial community. Conversely, Microvirga and Streptomyces, the dominant genera in Sorg/Pean soils, were negatively correlated with SOC. Besides the influence of SOC, soil TN, AN, and AP also played essential roles, and each genus was affected to a different degree by these abiotic factors. These results indicate that soil properties had a considerable effect on the structure of soil bacterial communities.

Tax4Fun analysis of 16S rRNA data could be applied as an inexpensive tool for bacterial functional prediction (Asshauer et al., 2015; Wang et al., 2020). In the present study, we found that different cropping systems shared similar metabolic pathways. Most of the predicted sequences were related to functions such as carbohydrate metabolism, amino acid metabolism, membrane transport, energy metabolism, metabolism of cofactors and vitamins, and signal transduction. Notably, the abundance of carbohydrate metabolism and amino acid metabolism functional profiles accounted for >24%. Carbohydrate metabolism integrates pathways that transport and oxidize primary carbon sources into the cell, which also plays a vital role in nitrogen and carbon removal (Papagianni, 2012; Chen et al., 2021). The amino acid metabolism can enhance microbial growth and activity by providing more amino acids as carbon and energy sources for microorganisms (Liang et al., 2020). Among the five treatments, bacterial community functions exhibited noticeable differences, indicating that cropping systems strongly impact bacterial composition and primary metabolic functions (Cheng, Chen & Zhang, 2019). However, since the inherent limitations of Tax4Fun, explanations for the predicted functions should be discreet (Cheng, Chen & Zhang, 2019). Therefore, an in-depth study is needed through metagenomic analyses to better understand the effects of cropping systems effect on the ecological function of the microbiome.

Intercropping has been widely adopted to enhance land productivity in sustainable farming systems worldwide, thereby decreasing the starvation of populations (Martin-Guay et al., 2018; Zhang et al., 2020a). In the current study, the LER values of the grain sorghum/peanut intercropping and millet/peanut intercropping were all greater than one, indicating that intercropping has advantages in terms of saving land resources. The changes in yield can partly reflect soil quality. Our results showed that two intercropping systems exhibited the advantage of increasing grain yield, indirectly indicating that intercropping could improve soil quality through different methods (Zhang et al., 2019). These changes might be related to changes in soil microbes and properties (Tao et al., 2017). PLS-PM showed that soil bacterial community had a direct effect on yield and indirectly affected yield by altering soil properties, proving that intercropping improves system yield by altering soil bacterial community.

Conclusions

Our results demonstrated that intercropping improved land utilization and system yield. The yield advantage of intercropping is mainly attributed to changes in soil bacterial communities and soil properties. Intercropping increased soil nutrients and promoted soil bacterial taxa involved in N cycling processes and plant promotion. Intercropping increased the relative abundance of predicted functions involved in soil bacterial community nutrient cycling and environmental adaption. This study helps further understand the influences of intercropping on soil bacterial community structure and diversity and provides a scientific basis for promoting intercropping in eastern China. Future work could be extended through metagenomic sequencing to analyze genes and metabolic pathways.

Supplemental Information

Figure S1 Field schematic diagrams of the sorghum/peanut 3:4 intercropping (A) and millet/peanut 4:4 intercropping (B)

Click here for additional data file.

Figure S2 Correlation analysis for soil properties

*p < 0.05, **p < 0.01, ***p < 0.001.

Click here for additional data file.

Figure S3 The number of sequencing reads of five cropping systems

Click here for additional data file.

Figure S4 Venn diagram showing the unique and shared ASVs among five cropping systems

Click here for additional data file.

Figure S5 Alpha diversity index of soil bacterial community under five cropping systems

(A) Shannon index; (B) chao1 index.

Click here for additional data file.

Figure S6 The relative abundance of predicted KEGG categories in the soils under the five cropping systems

*p < 0.05, **p < 0.01, ***p < 0.001.

Click here for additional data file.

Data S1 Raw data for the physical and chemical properties of the samples

Click here for additional data file.

The author thanks all members of the Ecological Laboratory of Shandong Academy of Agricultural Sciences for their help.

Additional Information and Declarations

Competing Interests

Author Contributions

Data Availability

The authors declare there are no competing interests.

Zhu Liu performed the experiments, analyzed the data, prepared figures and/or tables, and approved the final draft.

Zhenwu Nan performed the experiments, analyzed the data, prepared figures and/or tables, and approved the final draft.

Songming Lin performed the experiments, prepared figures and/or tables, and approved the final draft.

Weiwei Meng conceived and designed the experiments, prepared figures and/or tables, and approved the final draft.

Liyong Xie analyzed the data, authored or reviewed drafts of the article, and approved the final draft.

Haiqiu Yu analyzed the data, authored or reviewed drafts of the article, and approved the final draft.

Zheng Zhang conceived and designed the experiments, authored or reviewed drafts of the article, and approved the final draft.

Shubo Wan conceived and designed the experiments, authored or reviewed drafts of the article, and approved the final draft.

The following information was supplied regarding data availability:

The raw reads are available in the NCBI Sequence Read Archive (SRA): PRJNA1006275.

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
