# Peer review of "Peanut-based intercropping systems altered soil bacterial communities, potential functions, and crop yield"

_PeerJ, doi:10.7717/peerj.16907_

## Round 0.1 · original submission · Major Revisions

Both reviewers agreed that there were intriguing aspects to the data, but also some significant limitations, and so major revisions are required. I would also add the following comments:

1. Were sequencing libraries mixed with PhiX? If so, how were these reads removed?
2. How were sequencing adapter sequences removed from reads? This is usually done before analyzing with QIIME2.
2b. Was sequencing read quality observed before analyzing with QIIME2 (e.g. with fastqc or multiqc?)?
3. Was a positive control such as a mock microbial community used? To check if primers are more likely to amplify certain genera/don't amplify others.
4. Was a negative control used, so starting with the DNA extraction kit, to remove any potential contaminant reads?
4b. Can you be more specific about the Nextflex kit used? There's a Rapid XP DNA-seq kit, a DNA-seq 2.0 kit, a cell-free DNA-seq kit, etc.
4c. Barcodes used with this kit; do you re-use barcodes in combinations, or is each barcode only used for a single sample?
5. Consider adding ellipses to your NMDS plot, such as here: https://www.researchgate.net/figure/NMDS-showing-the-beta-diversity-of-A-Aedes-vexans-and-B-Culex-pipiens-microbiomes_fig3_339981588 (I'm not extolling virtues of the rest of the paper - I've not read it; I just googled for an example image).
6. Line 202: you report the total number of ASVs. Could you report the number of sequencing reads per sample and a standard deviation across the replicates for each treatment?

Reviewer 1 ·

Basic reporting

This paper examines the impacts of peanut-based intercropping on soil bacterial communities and crop yields, comparing it to monoculture. While the topic shows promise, the paper does have notable limitations and weaknesses that hinder its reliability and significance.
1) The field experiment conducted for only one year decreases the overall reliability of the research findings. Expanding the study duration could help enhance the robustness of the results.
2) Although the paper investigates the effects of peanut-based intercropping on soil properties and crop yields, it fails to establish a clear link between bacterial communities and crop productivity. Establishing this relationship would significantly strengthen the paper's findings.
3) The observed differences in bacterial communities between peanut-sorghum and peanut-millet intercropping are noteworthy, but further investigation is required to understand the underlying mechanisms and potential functions. This additional analysis would provide valuable insights into the microbial dynamics related to intercropping systems.
Overall, while the paper contains some intriguing data, it lacks the novelty, depth, and academic hypothesis that are necessary for it to be considered acceptable in its current form.

Experimental design

Good

Validity of the findings

Good

Additional comments

No

·

Basic reporting

The manuscript was written in professional English. There were a few grammatical errors and other major issues that need to be addressed to improve the readability of the manuscript.
Abstract:
Line 59: What is the broader impact of your study.
Introduction:
First paragraph: The introduction will read better if you start with the demerits of monocropping, then move to the merits of intercropping and why it is important. Thereafter, you can introduce why peanuts are considered in intercropping systems. Probably, because of its nitrogen fixing ability and other benefits listed in lines 70 – 73.
Line 89: Xue et al., 2023 did not work on maize and peanut. This citation is incorrect. They cited Chen et al. for this sentence. It is advisable that you read the direct authors that worked on maize/peanut system and cite them appropriately.
Last paragraph 99 -101: Please expand upon the knowledge gap being filled. Rewrite your objective and make it clear. What do inner mechanisms mean? Also, be specific on what you intend to use the Illumina Miseq platform to achieve. Did you carry out amplicon or metagenomic sequencing?
Discussion:
The discussion should not be a repetition of results. This is an avenue to discuss your findings, how your findings fill the knowledge gaps mentioned in the introduction, the limitations of your study and discuss the broader impacts of your study.

Experimental design

The experimental design (CRD or RCBD) was not mentioned in the manuscript. The data would have been more robust if replications were more than three, especially for a study set up in 2019. Every other aspect of the experimental design is okay.
Line 175: Please provide more information for sequence analysis. Did you use single or paired sequences for your analysis?

Validity of the findings

The statistical tests applied are appropriate. Conclusion lacks depth. It failed to state clearly the broader impacts of the study.

Additional comments

Other minor comments in the different sections
Minor issues
Abstract:
Line 43: change influences to influence
Line 44: There are many studies out there showing the effect of intercropping on soil bacterial communities. So, “rarely been reported” is untrue.
Line 51: This sentence, “The Actinobacteria, Proteobacteria, Acidobacteria, and Chloroflexi bacterial phyla were dominant across all cropping systems” can read better. I suggest writing it as : Bacterial phyla, including Actinobacteria, Proteobacteria ……, were dominant across all cropping systems.
Line 53: Spell enchanced correctly to enhanced. “Sorghum/peanut intercropping enchanced”

Introduction:
Line 76 -77: see my comments for line 44.
Line 84: Please specify the type of materials (organic?)
87: Soil or plant microbial communities?
88: delete both “the”
89: Does amount of soil bacteria mean relative or absolute abundance or diversity? If it is relative abundance, it is important to state it because relative abundance is different from absolute abundance.
Line 97: Clarify what you mean by high dwarf crops. Do you mean tall and dwarf crops?
Line 99: Do you mean that “more studies needed to understand the influence of intercropping on soil microbial communities need to be carried out in Shandong Province? If yes, please state it clearly.
Line 104: State the different types of intercropping systems investigated.

Materials and method:
Line 111 – 114. Cite the source of climate data.
Line 115: Please provide the soil pH.
Line 125 -126: Change “are” to “were”. Millet and sorghum were harvested in early Sept.
Line 187: Please provide a reference for LDA > 3.5
Line 188: Did you carry out a correlation analysis among soil physicochemical properties before selecting them for redundancy analysis? Strong and significant correlations among soil physicochemical properties can affect the RDA.

Results
Line 198: Quantify the increase. It could be quantified in percent
Line 204: Delete the phrase, regarding unique ASVs.
Line 210: delete “divided between” and replace with “among”
Line 221: “varies between treatments” should be “varied among treatments”
Line 250: move “further demonstrating that SOC could structure the bacterial community composition” to discussion.
Line 268: It would be nice to see the relationship between soil bacterial properties and total grain yield.

Figures:
Overall comment: The figure can be rearranged, starting with total grain yield and LER, soil chemical properties before proceeding with the soil bacterial components.
There are too many figures for the manuscript. I suggest that figures 1, 2, 3, and 8 to supplementary information.
Figure 3: remove the lowercase letters on the figure since there was no statistical difference.

Discussion:
Line 276: change “enhance” to “enhancing”
Line 278: Since your preceding sentence on line 276 talks about cereal/legume intercropping, it would be better to give an example of cereal/legume intercropping rather than using mulberry.
Line 279: Be specific with the two intercropping systems in your study. Name them
Line 282: Correct the grammar here. It should read: These results might be attributed to
Line 291 – 292: This sentence is not clear, how does competitiveness of resident microorganisms affect differences in microbial diversity.
Line 296: Based on your beta diversity, cropping systems affected the bacterial community composition. Or do you have a different definition for bacterial community composition? If so, please state it.
Line 298 – 299: This sentence doesn’t read clearly, please rephrase. An example: The dominant phyla include Actinobacateria, Proteobacteria ……., were consistent across all cropping systems, indicating that intercropping did not alter the dominant bacterial phyla.
This is just an example; you can rewrite yours to read better.
Line 301- 302: What is the implication of the result. You should discuss your result.
Line 315: How does increase in Streptomyces boost peanut growth and yield? Is it through the release of antimicrobial compounds or the mineralization of organic compounds to plant nutrients? Instead of repeating results in this section, it is important to discuss your results and the implications of your findings.
Line 315 – 322: You can rewrite this paragraph and discuss your result. For example: ‘Intercropping in our study promoted bacterial genera that are potentially involved in N fixation and cycling activities. For example, there was an increase in the relative abundance of Microvirga and Acidibacter, known for their roles in n fixation and xxxx, under Sorg/Pean soils. Similarly, we found an increase in Kribbella and Reyranella under Mill/Pean soils.”
Line 323 – 325: I like these sentences, they read nice.
Line 327: Edaphic factors affect microbes indirectly. pH can affect nutrient availability and indirectly affect soil microbes.
Line 332: Why are soil bacteria sensitive to SOC? Please explain further. Carbon serves as nutrient and energy sources.
Line 348: Can Tax4Fun identify the microbial groups linked to the metabolic pathways listed?

Conclusion:
The conclusion can be more detailed. State the contribution of your study to the body of knowledge. From your study, it is obvious that intercropping has several benefits from efficient land use, increasing soil nutrients, and promoting soil bacterial taxa involved in N cycling processes and plant promotion.

---

## Round 0.2 · Minor Revisions

While the reviewer was satisfied that you have addressed their concerns, there were still some statements in the methods sections that I felt could be clearer. I have added comments to your rebuttal letter, and attach these below.

·

Basic reporting

No comment

Experimental design

No comment

Validity of the findings

No comment

Additional comments

The authors addressed my concerns. I suggest that the authors read through the manuscript thoroughly to avoid grammatical errors that could be introduced during revisions.

---

## Round 0.3 · accepted · Accept

The methods section now more clearly describes the process followed, and figure S3 is showing the number of reads.